# Log-Augmented Generation: Scaling Test-Time Reasoning with Reusable Computation

**Peter Baile Chen**[1]    **Yi Zhang**[2]    **Dan Roth**[3]
**Samuel Madden**[1]    **Jacob Andreas**[1]    **Michael Cafarella**[1]
[1]MIT    [2]Greenshoe, Inc.    [3]Oracle AI and the University of Pennsylvania
Correspondence: `peterbc@mit.edu`

## Abstract

While humans naturally learn and adapt from past experiences, large language models (LLMs) and their agentic counterparts often fail to retain reasoning from previous tasks and apply it to future contexts. We introduce **L**og-**A**ugmented **G**eneration (LAG), a novel framework that *directly reuses* prior computation and reasoning from past logs at test time, enabling models to learn from previous tasks to perform better on new, unseen challenges, without sacrificing efficiency or scalability. Our approach represents task logs as key-value (KV) caches that encode the reasoning context of prior tasks, while storing KV values for only a selected subset of tokens. When a new task arises, LAG retrieves KV values from relevant logs to augment generation. Unlike reflection-based memory mechanisms, which require additional extraction or distillation steps, LAG reuses prior reasoning verbatim. Moreover, it extends beyond existing KV caching techniques, primarily designed for efficiency, by explicitly improving accuracy through log reuse. Experiments on knowledge- and reasoning-intensive datasets demonstrate that our method significantly outperforms standard agentic systems without log utilization, as well as existing approaches based on reflection and KV cache techniques.[1]

## 1 Introduction

The ability to learn from past experience is highly valuable, a skill that humans naturally possess but large language models (LLMs) do not have by default. Consider the two multi-hop questions shown in Figure 1, which share two intermediate reasoning steps. When humans tackle the first question, they naturally break it down into four steps, generate intermediate results, and remember them. Then, when faced with the second question, they can *reuse the relevant computations* from the first task, reducing the reasoning required from four steps to two steps. In contrast, when models process the two questions sequentially, they handle them independently without retaining memory of the previous task, which prevents them from recognizing the connection and reusing reasoning.

In this paper, we propose **Log-Augmented Generation** (LAG), a framework that directly reuses prior computation and reasoning from past logs at inference time to close this gap. To implement this framework, a natural idea is to store all model interactions as textual logs in a database or retrieval system and, when a new task arises, retrieve relevant logs to include in the model's context for generation. While full textual logs can retain rich contextual details, they often introduce substantial noise and may exceed context length limitations. We instead represent logs using KV values corresponding to a subset of tokens in past reasoning traces to represent the full reasoning context—reducing size while enabling context-dependent interpretation. Our method is inspired by the insight that the KV representations of individual tokens encapsulate far more meaning than the token alone: a token's KV value is a weighted aggregation of embeddings from the entire surrounding context, enabling it to capture the semantics of the broader sequence. This means it is possible to store only a subset of tokens along with their KV values, while still retaining the essence of the full reasoning context.

Our approach differs from prior work that relies on log extraction and reflection. Recent studies (Madaan et al., 2022; Shinn et al., 2023; Suzgun et al., 2025) propose methods that distill logs into

---

[1]Data and code are available at `https://peterbaile.github.io/lag/`.

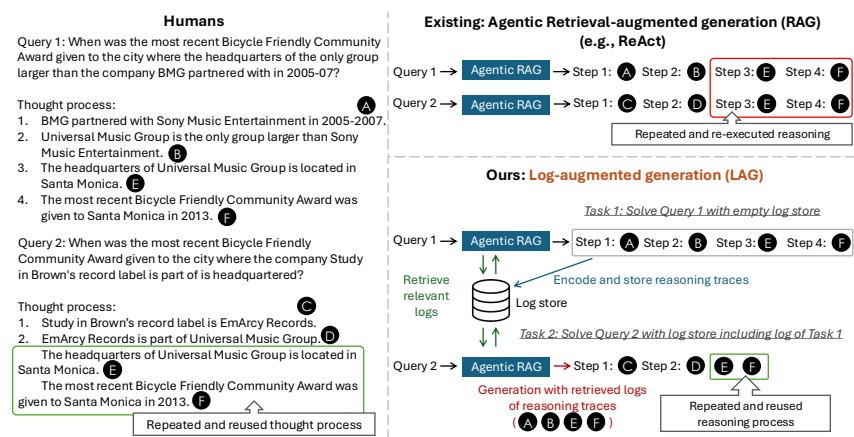

Figure 1: While humans naturally possess the ability to learn from past experiences, LLMs lack this capability, resulting in redundant reasoning. Our log-augmented generation framework allows LLMs to *directly* reuse prior thought/ reasoning processes. Algorithm 1 includes details of LAG.

reusable components like thoughts (Feng et al., 2024), problem-solving strategies (Yang et al., 2024), or domain-specific knowledge (Suzgun et al., 2025). However, such distilled content often lacks the full reasoning context and may become overly abstract, which can limit its practical usefulness and reusability. Instead, our framework *directly* encodes and *reuses* past reasoning and computation, avoiding the need for additional knowledge extraction and distillation steps. Our framework further utilizes KV representation of logs to implement this encoding for more effective reuse.

Our approach also stands apart from existing KV cache methods that focus on caching KV values for prompt instructions (Gim et al., 2024) or document corpora in retrieval-augmented generation (RAG) systems (Lu et al., 2024; Chan et al., 2024; Sun et al., 2024). In comparison, our system aims to cache and reuse the reasoning trace of previous executions. Moreover, the primary goal of these methods is to reduce redundant computation when identical content reappears. In contrast, we emphasize using KV caching not just for computational efficiency, but as a means to retain contextual information, enhancing not just efficiency but accuracy.

In summary, this paper introduces log-augmented generation, a new framework designed to enable the direct reuse of prior reasoning. LAG leverages KV values to represent past reasoning context to support effective reuse. We evaluated LAG on both knowledge-intensive tasks, such as open-domain multi-hop question answering; and reasoning-intensive tasks, including math and science questions. LAG significantly outperforms both standard agentic systems without log usage and existing reflection and KV caching techniques, achieving superior effectiveness and efficiency.

## 2 METHODOLOGY

We begin by outlining the general framework of log-augmented generation, which enables the direct reuse of prior reasoning, in Section 2.1. Next, in Section 2.2, we detail our specific implementation of the framework, focusing on how logs are represented using KV values.

### 2.1 LOG-AUGMENTED GENERATION

Our framework aims to enable LLMs to directly leverage previous executions when performing generative tasks in order to boost their performance on future tasks. This requires three additional components on top of the base model: (1) an encoding and storage module that represents and stores past executions in a form that can be easily and effectively reused; (2) a retrieval module that, given a new task, identifies the most relevant and useful prior history; (3) a generation module that integrates the retrieved prior history into the context of the LLM to enhance its generation.

Algorithm 1 outlines the general log-augmented generation framework applied to agentic systems, which involves multiple sequential LLM generations for multi-step reasoning. At the end of each step, the model chooses between two actions: either providing an answer to the task or generating a follow-up query or sub-task to progress toward a solution. This cycle continues until the task is resolved or, more realistically, until a set iteration limit is reached. Notably, our approach integrates seamlessly with such sequential generation workflows, requiring only minimal modifications: adding a storage step, a retrieval step, and slight adjustments to the prompting and model generation processes. Furthermore, while our framework is demonstrated within the context of agentic systems, it can be readily adapted to other generative paradigms, as the core loop of storing, retrieving, and augmented generation can be seamlessly integrated into any LLM-based generation process.

---

**Algorithm 1** Overview of our log-augmented generation framework (LAG) applied to agentic systems involving multiple sequential generations, with our framework generalizable to any generative tasks. Gray text indicates steps representing interaction with external tools (e.g., document retrieval) by the agentic system. Blue text shows additional steps introduced for LAG.

---

**Require:** Generator LM $\mathcal{M}$, number of maximum reasoning steps allowed $\mathcal{C}$, Document retriever $\mathcal{R}_{\mathcal{D}}$, Document store $\mathcal{D}$, Log encoder $\mathcal{E}$, Log retriever $\mathcal{R}_{\mathcal{L}}$, Log store $\mathcal{L}$

1: **Input:** input task $x$, **Output:** an answer
2: **Initialize:** model response $y$ to None, number of reasoning steps $c$ to 0, retrieved documents $\mathbf{D}$ to $\{\}$, retrieved logs $\mathbf{L}$ to $\{\}$, conversation history $\mathbf{H}$ to []
3: **while** $y$ does not include an answer and $c < \mathcal{C}$ **do**
4:     **if** $y$ is None **then**
5:         Next action $a$ is $x$
6:     **else**
7:         Extract next action $a$ from $y$
8:     Retrieve relevant documents $\mathbf{d}$ from $\mathcal{D}$ using $\mathcal{R}_{\mathcal{D}}$ given $a$ and update $\mathbf{D}$ with $\mathbf{d}$
9:     Retrieve relevant logs $\mathbf{l}$ from $\mathcal{L}$ using $\mathcal{R}_{\mathcal{L}}$ given $a$ and update $\mathbf{L}$ with $\mathbf{l}$
10:    Formulate user-defined prompt $p$ based on $x$, $a$, $y$, and $\mathbf{D}$. Append $\mathbf{L}$ to the beginning of $p$
11:    $\mathcal{M}$ generates $y$ given $p$ and $\mathbf{L}$
12:    Append $p$ and $y$ to $\mathbf{H}$
13: Update $\mathcal{L}$ with $\mathbf{H}$ using $\mathcal{E}$

---

**Log retrieval.** The retriever, $\mathcal{R}_{\mathcal{L}}$, selects relevant logs from $\mathcal{L}$ by performing standard semantic similarity ranking between stored logs and the next action $a$ (typically a follow-up query or sub-task). Specifically, $\mathcal{R}_{\mathcal{L}}$ retrieves the top-$k$ logs with the highest cosine similarity to $a$ based on their embeddings, which are generated using standard text embedding models. Log embeddings are precomputed and stored offline in $\mathcal{L}$, so at test time, only the embedding for $a$ needs to be computed.

The remaining two components—storage and generation—depend on the chosen log representation, which we will detail further in Section 2.2.

## 2.2 Logs represented as KV values

As outlined in Section 1, our goal is to represent logs in a way that retains the full reasoning context while keeping the amount of logs stored small, in order to reduce noise and avoid surpassing context length limits. While textual logs are a natural choice, capturing the entire reasoning context this way requires storing full reasoning traces, which can be inefficient and unwieldy. To address this, we propose using KV values as the log representation, leveraging the inherent property that the KV value of any token attends to the entire context. We first describe what information is encoded and stored in the log store $\mathcal{L}$, followed by a description of how the stored KV values are used to enhance model generation. The retrieval component $\mathcal{R}_{\mathcal{L}}$ was described above.

**KV values for context representation.** We achieve the effect of retaining the full reasoning context while storing a small number of tokens by differentiating between the content used as context for encoding the KV cache and the actual tokens stored within it. Crucially, the KV representation of a single token encapsulates more than just the token's meaning—it reflects the semantics of the entire context. KV values are produced via the attention mechanism. As shown in Figure 2, the attention weights allow each token to attend to all previous tokens, making a token's KV representation a weighted combination of the embeddings of all prior tokens in the sequence. Because of

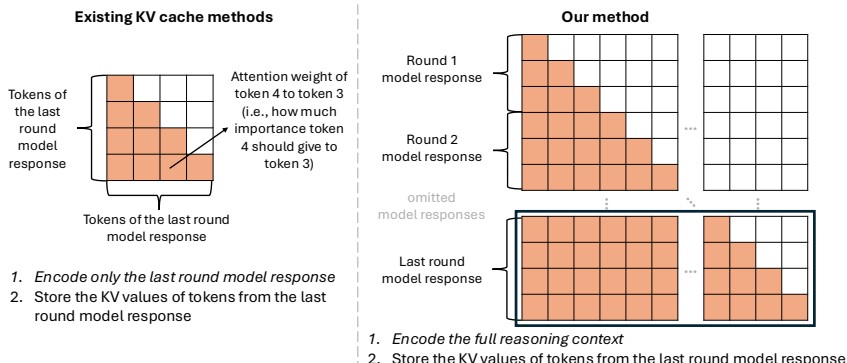

Figure 2: A typical attention weight, after applying the lower-triangular causal mask, enables each token's KV values to attend to all preceding context with different levels of importance. Leveraging this property, LAG selectively stores the KV values of a subset of tokens from the model's most recent response, while still preserving the complete reasoning context. This approach differs from existing KV caching methods, which do not differentiate between content for encoding and storage.

this property, storing only a subset of tokens and their associated KV values is sufficient to retain the essential semantics of the entire reasoning context.

As shown in Algorithm 1, a task includes multiple turns of user and assistant messages (we use "model response" and "assistant message" interchangeably), where the total number of turns depends on the specific generative framework, but should be at least one. User messages convey instructions, task details, and optional inputs such as retrieved content, while assistant messages represent the model's reasoning steps. To capture the full reasoning context, as illustrated in Figure 2, we encode a sequence of all model responses across all turns into KV values. For storage, however, we retain only the KV values associated with the last model response, treating it as a condensed summary of the entire reasoning trace. This last response is chosen because it likely reflects the model's most refined understanding of the task, allowing the corresponding KV values to attend to different parts of the overall reasoning context. This contrasts with existing KV cache methods, which do not distinguish between what gets encoded and what is stored (Lu et al., 2024; Sun et al., 2024). For example, when storing the KV cache of a document, these approaches typically encode and store the document's KV cache directly. A straightforward adaptation in our setting would be to treat the last model response independently—encoding and storing it as a KV cache without considering the reasoning from earlier responses, as illustrated in Figure 2.

**Token selection for KV values.** Although KV values encode richer semantics than plain text tokens, they come with a higher storage cost since each token's KV representation is a high-dimensional vector. To reduce storage costs, we may opt to retain the KV values for only a small subset of important tokens, such as those representing the final answer, or in agentic settings, the next action $a$ from Algorithm 1 when the answer cannot be determined—since these encapsulate the result of the reasoning process. Nonetheless, as we will discuss in Section 4.2, our analysis shows that storing KV values for a greater number of tokens generally improves performance, though at the expense of increased storage. Overall, we identify that storing KV values for the tokens in the last model response, our selected storage strategy, offers the best performance-storage tradeoff.

**Generation with KV values.** Once the relevant logs are identified via the retriever $\mathcal{R}_{\mathcal{L}}$, their corresponding KV values are retrieved and provided to the LLM to augment generation. However, simply concatenating them—analogous to concatenating multiple textual logs—and passing them to the LLM for generation is problematic, due to the dependence of KV values on positional information. The KV values in the log store $\mathcal{L}$ were generated based on the positional IDs within their original encoding context, which differ from those in the current context. Inspired by (Lu et al., 2024; Sun et al., 2024), we address this issue by first stripping the stored KV caches of their original positional embeddings, and then reapplying positional embeddings appropriate for their new con-

text. We present the basic concept below, which is a simplified representation of the actual matrix computations used in practice.

Current LLMs use Rotary Positional Embeddings (RoPE) to encode positional information by applying a position-dependent rotation matrix to each 2D subvector $x$ within the key vectors of the KV values, where the rotation matrix is determined by the rotation angle $\theta$ that is a function of the input token's positional ID.

$$\text{RoPE}(x, \theta) = \begin{bmatrix} \cos(\theta) & -\sin(\theta) \\ \sin(\theta) & \cos(\theta) \end{bmatrix} \cdot x \tag{1}$$

To remove the influence of positional encoding, we first multiply $\text{RoPE}(x, \theta)$ by the inverse of the rotary matrix to obtain $x$.

$$\begin{bmatrix} \cos(\theta) & \sin(\theta) \\ -\sin(\theta) & \cos(\theta) \end{bmatrix} \text{RoPE}(x, \theta) = \begin{bmatrix} \cos(\theta) & \sin(\theta) \\ -\sin(\theta) & \cos(\theta) \end{bmatrix} \begin{bmatrix} \cos(\theta) & -\sin(\theta) \\ \sin(\theta) & \cos(\theta) \end{bmatrix} x = x \tag{2}$$

Next, we compute a new rotary matrix using $\theta'$, which is derived from the updated positional IDs, allowing us to generate $\text{RoPE}(x, \theta')$ aligned with the new context.

$$\text{RoPE}(x, \theta') = \begin{bmatrix} \cos(\theta') & -\sin(\theta') \\ \sin(\theta') & \cos(\theta') \end{bmatrix} x \tag{3}$$

With the updated KV values for each log, we concatenate them and use them to augment LLM generation.

## 3 EXPERIMENTS

### 3.1 EXPERIMENTAL SETUP

**Datasets.** We selected both representative knowledge-intensive and reasoning-intensive datasets to assess the effectiveness of different methods. For the knowledge-intensive setting, we used open-domain multi-hop QA datasets: Musique (Trivedi et al., 2022) and 2WikiMultiHop (Ho et al., 2020). Specifically, we used the development split of Musique, which includes 2,417 questions, and a randomly sampled subset of 4,000 questions from the 2WikiMultiHop dev set. For reasoning-intensive tasks, we focused on math and science QA datasets: GPQA (Rein et al., 2024) and MMLU-Pro (Wang et al., 2024), using 448 questions from the GPQA development split and a random sample of 1,000 questions from the MMLU-Pro dev set.

**Metrics.** We use standard evaluation metrics to assess end-to-end performance on the different QA datasets. For short-answer tasks, such as those in multi-hop QA datasets, we report Exact Match (EM) and F1 scores by comparing the predicted and gold answers. For multiple-choice questions in the math and science QA datasets, we evaluate using exact match between the predicted and correct choices. Additionally, we report the average number of reasoning iterations required to solve each task as a measure of efficiency.

**Baselines.** We evaluated our approach by comparing it with problem-solving frameworks that do not utilize logs, existing methods based on reflection and KV cache techniques, and an ablated version of our log-augmented generation method where logs are represented as plain text.

- Standard agentic: We employed a ReAct-style agentic reasoning framework (Yao et al., 2023) as a strong problem-solving framework, given its retry capability and adaptability across diverse task types. As discussed in Section 2.1, the agentic system operates iteratively, deciding at each step whether to provide an answer or initiate a new reasoning action, which may include interactions with external tools.

- Reflection: As a representative baseline for reflection-based systems, we included Dynamic Cheatsheet (Suzgun et al., 2025). This method enhances generation through online reflection, retrieving the top-$k$ most relevant logs and extracting key insights from them that are applicable to the current user task. The insights are subsequently incorporated into the model's context to enhance its generation.

- KV cache: We adopt Block Attention (Sun et al., 2024) as a representative baseline for existing KV cache methods. As discussed in Section 2.2, this approach does not distinguish between the content being encoded and the content stored as KV values. Specifically, we implement the version that encodes the last model response and stores the KV values corresponding to the tokens in that response. During generation, it retrieves the top-$k$ most relevant logs in their KV format, which are then used to enhance the model's generation process.

- $\text{LAG}_{\text{text}}$: To assess the impact of different log representations, we introduce a variant of our log-augmented generation framework that is identical in all aspects except for the log representation—it uses texts instead of KV values. We implemented two versions of $\text{LAG}_{\text{text}}$. $\text{LAG}_{\text{text}}$-all refers to storing and augmenting models with reasoning traces from *all* rounds. $\text{LAG}_{\text{text}}$-last refers to storing and augmenting models with reasoning traces from the *last* round only.

All baselines that utilize logs are implemented on top of the standard agentic framework for generation. For retrieving relevant logs, $\text{LAG}_{\text{text}}$-all stores the full reasoning trace, so the retriever compares the input query with the entire trace. In contrast, all other baselines store only the final round of reasoning, so the retriever uses just that last round to compute semantic similarity. In this section, we denote the default version of our log-augmented generation framework that represents logs using KV values as $\text{LAG}_{\text{KV}}$, to distinguish it from the variant of our framework that utilizes textual representations.

**Environment.** We used several top-performing lightweight embedding models from the MTEB leaderboard (Muennighoff et al., 2022), including `Snowflake-arctic-embed-m-v2.0` (Yu et al., 2024) and `Qwen3-Embedding-0.6B` (Zhang et al., 2025). For method execution, we used open-source LLMs that allow direct access to and manipulation of KV values, specifically `Llama-3.1-8B-Instruct`, `Qwen3-30B-A3B-Instruct-2507`, and `Llama-3.1-70B-Instruct`. The main paper reports results using `Llama-3.1-8B` with `Snowflake`, while additional evaluations with other embedding models and LLMs are presented in Appendix E and Appendix F, respectively. The prompts used for ReAct-style agentic systems are provided in Appendix A. Since agentic systems perform iterative reasoning until an answer is found, and this process generally lacks a convergence guarantee, it is necessary to set a maximum number of reasoning steps. For multi-hop question answering, we cap the number of reasoning steps at 8, as these tasks usually require up to 4 hops. For reasoning-intensive tasks, we limit the number of steps to 3. In all settings that have access to logs, we retrieve top-3 most relevant logs, and in RAG setups, we additionally retrieve top-2 most relevant documents.

We evaluate methods under two log store settings, static and dynamic, which differ in when logs are indexed. In the static log store setting, the store is constructed offline: we assume a predefined set of tasks has been executed in advance to generate logs, which are then indexed into the log store. During online usage, this fixed store can be queried for relevant logs, but remains unchanged until the next offline update. In contrast, a dynamic log store continuously evolves: after each task completes, its newly generated logs are immediately added and indexed. The main paper reports results for the static log store setting, while results for the dynamic log store setting are included in Appendix G. Under the static log store setting, we split each dataset into X% for constructing the log store and Y% for evaluating on "unseen" test questions. The X% portion used to populate the log store is referred to as the "seen" questions. To reflect realistic deployment conditions where ground truth answers are typically unavailable, we do not use gold answers to filter out incorrect logs during the log storage phase. However, developers who have access to ground truth data may choose to apply such filtering to improve log quality and overall system performance. We evaluated the robustness of our approach across multiple data splits (X/Y = 70/30, 50/50, and 30/70). The main paper reports results with X = 70 and Y = 30, while results for the other splits are shown in Appendix H.

## 3.2 RESULTS

The accuracy (measured as exact match and F1 scores) and efficiency (number of reasoning iterations) numbers are reported in Tables 1 and 2. In general, $\text{LAG}_{\text{KV}}$ outperforms all baselines across both knowledge- and reasoning-intensive datasets in terms of both metrics. Notably, the variant of

Table 1: Exact Match (EM) and F1 scores on knowledge-intensive datasets Musique and 2WikiMultiHop, along with the number of iterations required to complete each task. Seen or unseen refers to whether the questions were used to construct log stores. **Bolded** numbers indicate the best performance among all methods, and Underlined numbers indicate the performance of $LAG_{text}$ or $LAG_{KV}$ when they rank second.

| | Musique (seen) | | | Musique (unseen) | | | 2WikiMultiHop (seen) | | | 2WikiMultiHop (unseen) | | |
|---|---|---|---|---|---|---|---|---|---|---|---|---|
| | EM | F1 | #Iter. ↓ | EM | F1 | #Iter. | EM | F1 | #Iter. | EM | F1 | #Iter. |
| Standard agentic (Yao et al., 2023) | 26.3 | 36.4 | 3.98 | 27.0 | 37.3 | 3.90 | 50.2 | 58.9 | 2.53 | 51.6 | 60.2 | 2.49 |
| Reflection (Suzgun et al., 2025) | 26.6 | 37.5 | 3.58 | 27.5 | 38.8 | 3.67 | 48.3 | 57.1 | 2.17 | 50.1 | 59.2 | 2.22 |
| KV cache (Sun et al., 2024) | 28.5 | 38.5 | 3.36 | 28.7 | 40.7 | 3.37 | 52.9 | 61.1 | 1.93 | 48.3 | 57.1 | 1.89 |
| $LAG_{text}$-all | 26.4 | 35.0 | 4.34 | 27.8 | 37.1 | 4.30 | **57.5** | **65.6** | 2.52 | **56.9** | **65.0** | 2.29 |
| $LAG_{text}$-last | 29.6 | 40.0 | 3.17 | 30.7 | 42.1 | 3.20 | 55.8 | 64.1 | **1.77** | 55.0 | 63.6 | 2.00 |
| $LAG_{KV}$ | **32.3** | **43.2** | **2.65** | **32.2** | **45.0** | **2.68** | **57.5** | **66.1** | **1.77** | 55.2 | 64.7 | **1.84** |

Table 2: Performance on reasoning-intensive datasets GPQA and MMLU-Pro.

| | GPQA (seen) | | GPQA (unseen) | | MMLU-Pro (seen) | | MMLU-Pro (unseen) | |
|---|---|---|---|---|---|---|---|---|
| | EM | #Iter. ↓ | EM | #Iter. | EM | #Iter. | EM | #Iter. |
| Standard agentic Yao et al. (2023) | 22.4 | 1.96 | 18.5 | 1.87 | 39.7 | 1.68 | 41.3 | 1.58 |
| Reflection Suzgun et al. (2025) | 21.4 | 1.81 | 20.0 | 1.84 | 40.0 | 1.54 | 41.0 | 1.59 |
| KV cache Sun et al. (2024) | 19.5 | 1.91 | 19.3 | 1.68 | 37.3 | 1.61 | 40.7 | 1.51 |
| $LAG_{text}$-all | 22.4 | 1.94 | 17.0 | 1.92 | 39.7 | 1.62 | 39.7 | 1.60 |
| $LAG_{text}$-last | 21.7 | 1.89 | 18.5 | 1.93 | 41.6 | 1.53 | 42.0 | 1.50 |
| $LAG_{KV}$ | **24.6** | **1.58** | **30.4** | **1.62** | **44.3** | **1.33** | **42.3** | **1.36** |

our log-augmented generation framework that uses plain text ($LAG_{text}$) to represent logs generally ranks as the next most effective approach.

**Benefits of log-augmented generation.** Both $LAG_{KV}$ and $LAG_{text}$ surpass standard agentic systems that do not leverage logs, underscoring the value of logs in improving both the accuracy and efficiency of these systems. Furthermore, their superior performance over reflection-based methods suggests that directly reusing past reasoning and computation steps is more effective than approaches that rely on knowledge extraction and abstraction, which might produce overly abstract representations that lose the richness of the original reasoning context.

**Benefits of KV values for log representation.** The superior performance of $LAG_{KV}$ over $LAG_{text}$ demonstrates that KV representations are more effective than plain text for capturing and utilizing logs. When $LAG_{text}$ include all rounds of reasoning—matching what is encoded in the KV values of $LAG_{KV}$—they perform worse, likely because the full reasoning trace contains noise, whereas the KV values produced by the attention mechanism may emphasize more pertinent aspects. On the other hand, when $LAG_{text}$ include only the last round—matching what $LAG_{KV}$ stores—they also underperform, likely because they lose the broader reasoning context, which KV representations are still able to capture through encoding the full reasoning process.

Moreover, the fact that $LAG_{KV}$ outperforms existing KV cache methods—which do not differentiate between what is encoded and what is stored—indicates that the encoding strategy used in these methods fails to capture the full reasoning context. In contrast, $LAG_{KV}$ effectively preserves this broader context by encoding the entire reasoning context. We also note that adding more content for encoding does not affect the resulting KV size, as the size is determined solely by the number of stored tokens, the number of attention heads, and the dimension of each attention head—the latter two being fixed by the model architecture. Therefore, our more effective encoding strategy offers a performance gain without incurring additional storage cost.

## 4 ANALYSIS

Section 3.2 demonstrated that LAG outperforms existing approaches. In this section, we present a deeper analysis to better understand the sources of this improvement (Section 4.1). We also examine the tradeoff between performance and storage when representing logs with different token subsets

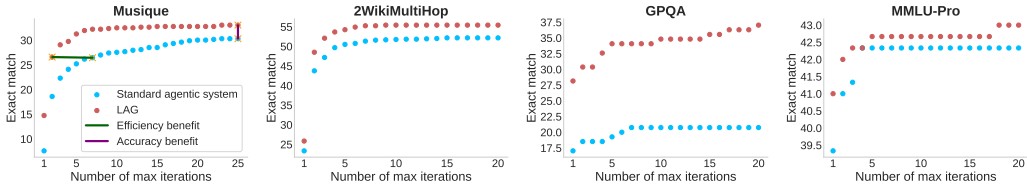

Figure 3: Exact match score of standard agentic systems and LAG varying the maximum number of reasoning steps allowed to solve a task. Logs show efficiency benefits when the performance of LAG in on par with agentic systems but requires fewer reasoning steps. They show performance benefits when LAG surpasses the agentic systems at points where their performance has plateaued.

Table 3: Breakdown of accuracy improvements at iteration 20 (25 for Musique), the point where both agentic systems and LAG reach a performance plateau. I, C, and U represent incorrect, correct, and unsolvable, respectively. X → Y refers to the number of questions that were initially X but became Y after incorporating logs.

|  | #Correct ans. w/o logs | I → C | C → I | U → C | C → U | Total improvement |
|---|---|---|---|---|---|---|
| Musique | 220 | +64 | -56 | +15 | -3 | +20 |
| 2WikiMultiHop | 626 | +138 | -111 | +12 | -0 | +39 |
| GPQA | 28 | +18 | -10 | +15 | -1 | +22 |
| MMLU-Pro | 127 | +35 | -39 | +12 | -6 | +2 |

(Section 4.2), and evaluate how performance varies with the number of retrieved logs (Section 4.3). All analyses were conducted on unseen questions across all datasets.

## 4.1 ACCURACY AND EFFICIENCY BENEFITS OF LOGS

Section 3.2 highlights how LAG enhances both the accuracy and efficiency of standard agentic systems. In this section, we take a closer look to better understand why logs are effective. Logs of past reasoning trace, intuitively, should support both knowledge reuse and insight reuse. Knowledge reuse allows partial solutions from past tasks to be applied to new ones, enabling the model to solve tasks using fewer reasoning steps and thus improving efficiency, which is illustrated in Figure 1. Insight reuse, on the other hand, helps the model identify prior mistakes, leading to correct answers where it previously failed, and recognize successful problem-solving strategies, allowing it to answer previously unsolvable questions, ultimately enhancing accuracy.

We define "unsolvable questions" as those that agentic systems are unable to answer within a reasonable number of reasoning steps. As discussed in Section 3.1, although agentic systems can theoretically continue reasoning indefinitely until a solution is found, this is not practical. To address this, we set an upper limit on the number of iterations. We extend this by increasing the maximum allowed iterations until the accuracy of standard agentic systems levels off. At this plateau, any questions that remain unanswered are considered unsolvable. As shown in Figure 3, our experiments show that performance flattens at around 20-25 iterations across all datasets. The figures also include the accuracy of LAG across different maximum iteration limits.

As shown in Figure 3, we observe clear efficiency and accuracy gains by LAG. For example, in the Musique dataset, LAG reaches an exact match score at iteration 2 that is slightly higher than that of standard agentic systems at iteration 7, yielding a 3.5x efficiency improvement. Additionally, even at iteration 25, where both systems reach their peak performance, LAG still outperforms the standard agentic system, demonstrating superior accuracy. To further illustrate this, Table 3 breaks down the sources of this accuracy improvement across all datasets. The results indicate that while the model may be misled by relevant log information and therefore turns some previously correct questions into incorrect or unsolvable ones, they more often convert previously incorrect or unsolvable questions into correct ones, highlighting their overall effectiveness in improving accuracy.

Table 4: Performance of LAG$_{KV}$ (using KV representation) and the size of the stored KV values (in GB) varying the content stored as KV values, along with the performance of standard agentic systems and the variant of our log-augmented generation framework that represents logs as texts (all reasoning traces stored and provided), LAG$_{text}$-all.

| | Musique | | | | 2WikiMultiHop | | | | GPQA | | | | MMLU-Pro | | | |
|---|---|---|---|---|---|---|---|---|---|---|---|---|---|---|---|---|
| | EM | F1 | #Iter. ↓ | KV size (GB) ↓ | EM | F1 | #Iter. | KV size | EM | F1 | #Iter. | KV size | EM | F1 | #Iter. | KV size |
| Standard agentic | 27.0 | 37.3 | 3.90 | - | 51.6 | 60.2 | 2.49 | - | 18.5 | - | 1.87 | - | 41.3 | - | 1.58 | - |
| LAG$_{text}$-all | 27.8 | 37.1 | 4.30 | - | 56.9 | 65.0 | 2.29 | - | 17.0 | - | 1.92 | - | 39.7 | - | 1.60 | - |
| LAG$_{KV}$ | | | | | | | | | | | | | | | | |
| Last 3 rounds | 33.2 | 45.1 | 3.50 | 93 | 57.2 | 66.0 | 1.98 | 91 | 20.7 | - | 2.03 | 57 | 34.3 | - | 1.57 | 88 |
| Last 2 rounds | 33.6 | 45.4 | 3.22 | 70 | 56.4 | 66.4 | 1.94 | 76 | 22.2 | - | 1.81 | 44 | 38.3 | - | 1.51 | 70 |
| Last round | 32.2 | 45.0 | 2.68 | 38 | 55.2 | 64.7 | 1.84 | 44 | 30.4 | - | 1.62 | 27 | 42.3 | - | 1.36 | 46 |
| Last action | 29.3 | 40.3 | 3.55 | 6.1 | 56.2 | 64.0 | 2.34 | 5.0 | 25.9 | - | 1.84 | 7.0 | 42.0 | - | 1.60 | 7.3 |

**Qualitative examples.**   We manually reviewed questions and selected representative examples that illustrate the reuse of knowledge and insights, which contributes to improving the accuracy and efficiency of standard agentic systems. Additional details are provided in Appendix B.

**Lightweight safeguards.**   Table 3 shows that logs can sometimes introduce errors, motivating the use of two lightweight and easy-to-apply safeguards: (1) similarity thresholding, which filters out low-relevance logs during retrieval, and (2) a brief reflection step, which prompts the model to assess the trustworthiness of the retrieved information before using it. These mechanisms act as fallback strategies that enhance the robustness of LAG. Detailed results are provided in Appendix I.

## 4.2   KV VALUES FOR CONTEXT REPRESENTATION

As mentioned in Section 2.2, although KV values are a more effective representation than text, they incur higher storage costs due to each token's KV being a high-dimensional vector. To address this, we explore how varying the subset of tokens whose KV values are stored impacts performance—while still encoding the full reasoning trace. Our goal is to understand how different levels of stored content affect the tradeoff between model performance and storage efficiency in LAG. Storing fewer tokens leads to a higher "compression" rate, lowering storage requirements but potentially reducing the fidelity of the contextual representation.

Table 4 presents the performance and storage footprint associated with different token storage strategies, alongside baselines that either provide no logs or represent all past reasoning traces as plain text. The "last action" strategy refers to storing only the tokens tied to the model's final action—either an answer or a follow-up question. Overall, we find that storing the last model response yields the best balance, achieving superior accuracy and efficiency while maintaining a reasonable storage cost. Additionally, we observe that storing only the KV values corresponding to the model's last action performs comparably to storing the entire last model response. This approach still outperforms standard agentic systems without logs and the textual log representation that includes all model responses. Importantly, it achieves this while significantly reducing the average KV storage size across all datasets, by a factor of 6.30, bringing the average down to 6.35GB, making it a practical and efficient tradeoff for developers who aim to optimize for storage.

We observe that selecting tokens corresponding to the last action may be effective because the final action represents the outcome of the reasoning process. While it may consist of only a few tokens, the corresponding KV values likely encapsulate and attend to a broader span of the earlier reasoning steps. These findings highlight the strength of KV as a representation, capable of encoding richer semantic information than plain text tokens while requiring significantly fewer tokens, thereby keeping storage costs relatively reasonable.

## 4.3   IMPACT OF THE NUMBER OF LOGS RETRIEVED ON PERFORMANCE

We also investigate how the number of retrieved logs affects the overall performance. Table 5 presents the results of LAG as the number of retrieved logs ($k$) varies from 0 to 3, where $k = 0$ represents the baseline standard agentic system. Overall, we find that increasing the number of retrieved logs generally leads to improved performance.

Table 5: Exact Match (EM) and F1 scores of LAG varying the number of logs $k$ retrieved at each reasoning step. $k = 0$ refers to standard agentic systems.

| | $k = 0$ | | | $k = 1$ | | | $k = 2$ | | | $k = 3$ | | |
|---|---|---|---|---|---|---|---|---|---|---|---|---|
| | EM | F1 | #Iter. ↓ | EM | F1 | #Iter. ↓ | EM | F1 | #Iter. ↓ | EM | F1 | #Iter. ↓ |
| Musique | 27.0 | 37.3 | 3.90 | 29.5 | 41.3 | 2.84 | 32.2 | 44.5 | 2.76 | 32.2 | 45.0 | 2.68 |
| 2WikiMultiHop | 51.6 | 60.2 | 2.49 | 53.7 | 61.8 | 2.05 | 54.4 | 63.6 | 1.91 | 55.2 | 64.7 | 1.84 |
| GPQA | 18.5 | - | 1.87 | 23.0 | - | 1.73 | 30.4 | - | 1.61 | 30.4 | - | 1.62 |
| MMLU-Pro | 41.3 | - | 1.58 | 41.0 | - | 1.42 | 39.0 | - | 1.40 | 42.3 | - | 1.36 |

## 5 CONCLUSION

This work presents LAG, a new framework that equips LLMs with the ability to directly reuse reasoning traces from previous tasks. Central to this framework is the use of KV values to represent logs, preserving essential reasoning context more effectively than plain text, while keeping storage requirements manageable. Experiments across both knowledge- and reasoning-intensive datasets demonstrate that LAG substantially outperforms standard agentic systems and surpasses existing reflection-based approaches and KV caching techniques.

ACKNOWLEDGMENTS

We thank Yongjoo Park and Tim Kraska for their helpful discussions. We gratefully acknowledge support from the DARPA ASKEM Award HR00112220042, the ARPA-H Biomedical Data Fabric project, Liberty Mutual, Jane Street, the MIT MGAIC Consortium, the MIT Everest Projects, the MIT-IBM Watson AI Lab, ONR-LtRD (N00014-23-1-2364), and the Google PhD Fellowship in collaboration with MIT. We also thank Amazon, Google, and Intel for their contributions through the MIT Data Systems and AI Lab (DSAIL).

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

## A    PROMPTS

The prompts used in the agentic system are shown in Tables 6 and 7.

Table 6: Prompt used for knowledge-intensive datasets: Musique and 2WikiMultiHop. Blue texts indicate changes introduced for LAG.

---

{retrieved logs}

Do not use your general knowledge. Do not assume the existence of external knowledge. Do not make any guesses.

You are provided with a user question, and information that might be relevant to the user question.

Your task consists of the following steps:
1. From the provided information, extract facts that is relevant to the user question

2. Based on the provided information only, determine if you have sufficient information to answer the user question
- If you can determine the answer, output a short answer (in a few words) to the user question. The short answer must be wrapped in <ans></ans>.
- If you cannot determine the answer, output some keywords that can help you retrieve new information. The keywords must be wrapped in <keywords></keywords>.

Here is the information:
{retrieved documents}

Here is the user question:
{Question}

---

Table 7: Prompt used for reasoning-intensive datasets: GPQA and MMLU-Pro. Blue texts indicate changes introduced for LAG.

{retrieved logs}

You are provided with a multi-choice question. Your task consists of the following steps:

1. From the provided information, extracts the key insights helpful for solving the user question

2. Break down and solve the question step by step, without relying on the provided answer choices

3. Based on your analysis, determine if you have sufficient information to identify the single most probable answer

- If you can identify the answer, output the answer as the letter corresponding to the answer choice, placed inside parentheses and wrapped in <ans></ans> (e.g., <ans>(A)</ans>).

- If you cannot identify the answer, output sub-questions that, if solved, can lead to new information. The sub-questions must be wrapped in <subquestion></subquestion>.

Here is the information:
{previous round model response}

Here is the user question:
{Question}

Here are the multiple-choice answers:
{Multiple choices}

## B  EXAMPLES WHERE LAG ENHANCES STANDARD AGENTIC SYSTEMS

Tables 8 and 9 present an example from the multi-hop QA dataset Musique, illustrating how LAG retrieves logs of previous questions containing *partial solutions* that can be directly applied to new questions. Our method reduces the number of reasoning rounds needed—from 9 to just 1—and corrects an originally incorrect answer.

Tables 10 and 11 present an example from the science QA dataset GPQA, where LAG retrieves logs from prior questions that contain reusable *insights*. These insights help the model recognize a previously overlooked aspect of the problem, allowing it to correct an answer that was initially incorrect.

Table 8:  At round 9, the standard agentic system produced an incorrect response. It failed to extract the **"role" information** (highlighted in bold) from the document, and returned the producer's name without identifying the role he plays.

---

**User message**

(...omitted instructions and passages...)

Document title: The Godfather Part II
Document content: The Godfather Part II is a 1974 American crime film produced and directed by Francis Ford Coppola from a screenplay co-written with Mario Puzo, starring Al Pacino and Robert De Niro. Partially based on Puzo's 1969 novel The Godfather, the film is both sequel and prequel to The Godfather, presenting parallel dramas: one picks up the 1958 story of Michael Corleone (Pacino), the new Don of the Corleone crime family, protecting the family business in the aftermath of an attempt on his life; the prequel covers the journey of his father, **Vito Corleone (De Niro)**, from his Sicilian childhood to the founding of his family enterprise in New York City.

Document title: Mistress (1992 film)
Document content: Mistress is a 1992 comedy-drama film starring Robert De Niro, Danny Aiello, Eli Wallach, Robert Wuhl and Martin Landau. The picture was written by Barry Primus and J.F. Lawton and directed by Primus.

(...8 documents omitted...)

Here is the user question: In The Godfather Part II, who did the producer of Mistress play?

---

**Assistant message (Round 9)**

(...omitted reasoning steps...)
Since Robert De Niro played a role in The Godfather Part II, and he is also a star in the film Mistress, we can determine that the answer is related to Robert De Niro.
- However, we **cannot determine which specific role Robert De Niro played in The Godfather Part II** from the provided information.
Therefore, the answer is: <ans>Robert De Niro</ans>

---

Table 9: In contrast to the standard agentic system's response shown in Table 8, LAG enables models to directly reuse knowledge embedded in prior reasoning traces, eliminating the need to retrieve and analyze relevant documents. This results in an 8-round reduction in problem-solving and yields a correct answer. **The reusable knowledge** is bolded.

---

**User message (retrieved logs included at the beginning)**

(...omitted logs...)

(past reasoning trace corresponding to the question *Who was played by the director of The Good Shepherd in The Godfather Part II?*)
(...omitted reasoning trace...)
- The director of The Good Shepherd is Robert De Niro.
- **In The Godfather Part II, Vito Corleone was played by Robert De Niro.**
(...omitted reasoning trace...)

(...omitted logs...)

- - - - - - - - - - - - - - - - - - - - - - - - - - - - - - - - - - - - - - - - - - -

(User prompt)
(...omitted instructions and passages...)

Document title: Mistress (1992 film)
Document content: Mistress is a 1992 comedy-drama film starring Robert De Niro, Danny Aiello, Eli Wallach, Robert Wuhl and Martin Landau. The picture was written by Barry Primus and J.F. Lawton and directed by Primus.

Here is the user question: In The Godfather Part II, who did the producer of Mistress play?

---

**Assistant message (Round 1)**

(...omitted reasoning steps...)
- **We have the name of the actor who played in The Godfather Part II (Robert De Niro).**
- **We have the name of the film he played in (The Godfather Part II).**
- We have the name of the film he played in (Mistress).
- We have the name of the producer of Mistress (Barry Primus).
Since we have the name of the actor, the film he played in, and the name of the producer of Mistress, we can determine the answer. <ans>Vito Corleone</ans>

---

Table 10: An incorrect response from the standard agentic system, which ignored E/Z-configurable double bonds, which also contributes to stereoisomer count.

---

**User message**

(...omitted instructions...)
How many stereoisomers are there for the compound 6-chloro-9-ethyl-2-methylundeca-3,7-dien-5-ol?
(...omitted instructions...)

---

**Assistant message (Round 1)**

To solve this problem, we need to understand the concept of stereoisomers and how to calculate them for a given compound.
1. **Key Insights:**
- Stereoisomers are molecules that have the same molecular formula and sequence of bonded atoms but differ in the three-dimensional orientations of their atoms in space.
- The number of stereoisomers can be calculated based on the number of chiral centers (asymmetric carbon atoms) in the molecule.
- Each chiral center can have two possible configurations (R or S), leading to $2^n$ possible stereoisomers, where n is the number of chiral centers.

2. **Breaking Down the Problem:**
- The given compound is 6-chloro-9-ethyl-2-methylundeca-3,7-dien-5-ol.
- To find the number of stereoisomers, we need to identify the chiral centers in the molecule.
- A chiral center is an asymmetric carbon atom with four different groups attached to it.
- We will analyze the molecule to identify the chiral centers.

3. **Analyzing the Molecule:**
- The molecule has the following structure: 6-chloro-9-ethyl-2-methylundeca-3,7-dien-5-ol.
- We can break down the molecule into its components:
- The main chain has 11 carbon atoms.
- There are two double bonds at positions 3 and 7.
- There is a chlorine atom at position 6.
- There is an ethyl group at position 9.
- There is a methyl group at position 2.
- There is a hydroxyl group at position 5.
- Based on the structure, we can identify the chiral centers:
- The carbon atom at position 2 has a methyl group, a hydrogen atom, a double bond, and a hydroxyl group attached to it, making it a chiral center.
- The carbon atom at position 5 has a hydroxyl group, a hydrogen atom, a double bond, and a chlorine atom attached to it, making it a chiral center.
- The carbon atom at position 9 has an ethyl group, a hydrogen atom, a double bond, and a chlorine atom attached to it, making it a chiral center.

4. **Calculating the Number of Stereoisomers:**
- We have identified three chiral centers in the molecule.
- Each chiral center can have two possible configurations (R or S).
- Therefore, the total number of stereoisomers is $2^3 = 8$.

Based on our analysis, we can conclude that the number of stereoisomers for the compound 6-chloro-9-ethyl-2-methylundeca-3,7-dien-5-ol is 8. <ans>(D)</ans>

---

**Gold reasoning from human expert**

Considering **each double bond** as a stereocenter, we have a total of four stereocenters, and thus it will have 16 stereoisomers (using the formula $2^n$, where n is the number of stereocenters).

Table 11: Unlike the standard agentic system's response in Table 10, LAG enables the model to uncover **insights** (bolded) embedded in prior reasoning steps—particularly, the relationship between E/Z-configurable double bonds and chiral centers. Although this insight doesn't directly solve the problem, it helps guide the model to use double bonds as a basis for determining the number of stereoisomers, a detail that was completely missed in the earlier assistant response in Table 10. This enables the model to arrive at the correct answer.

---

**User message (retrieved logs included at the beginning)**

(past reasoning trace corresponding to the question *How many of the following compounds will exhibit optical activity?*)
To solve this question, we need to understand what optical activity is and which compounds exhibit it. Optical activity is a property of a compound that can rotate plane-polarized light. This property is typically associated with chiral molecules, which have a non-superimposable mirror image.
Let's break down the given compounds and identify their chirality:
1. (Z)-1-chloro-2-methylbut-1-ene: This compound is an alkene with a (**Z**) **configuration**, which means it has a specific arrangement of atoms but is not **chiral**...

(...omitted logs...)

- - - - - - - - - - - - - - - - - - - - - - - - - - - - - - - - - - - - - - - - - - - - -

(User prompt)
(...omitted instructions...)
How many stereoisomers are there for the compound 6-chloro-9-ethyl-2-methylundeca-3,7-dien-5-ol?
(...omitted instructions...)

---

**Assistant message (Round 1)**

To solve this question, let's break it down step by step.
1. The compound 6-chloro-9-ethyl-2-methylundeca-3,7-dien-5-ol has a complex structure with multiple double bonds and a hydroxyl group. This suggests that it may have multiple stereocenters.
2. **The compound has three double bonds (3, 7, and 5), which means it has the potential for multiple stereoisomers due to the possibility of E/Z isomerism.**
3. The compound also has a hydroxyl group (5-ol), which means it has a stereocenter at the 5-position.
4. To determine the number of stereoisomers, we need to consider the number of stereocenters and the possible configurations at each stereocenter.
5. The compound has 3 **double bonds**, which can result in $2^3 = 8$ possible **E/Z configurations**.
6. The compound has 1 stereocenter (5-ol), which can have 2 possible configurations (R and S).
7. To calculate the total number of stereoisomers, we need to multiply the number of E/Z configurations by the number of possible configurations at the stereocenter. 8. The total number of stereoisomers is 8 (E/Z configurations) x 2 (possible configurations at the stereocenter) = 16.
Based on the analysis, we have sufficient information to identify the single most probable answer. <ans>(C)</ans>

---

**Gold reasoning from human expert**

Considering **each double bond** as a stereocenter, we have a total of four stereocenters, and thus it will have 16 stereoisomers (using the formula $2^n$, where n is the number of stereocenters).

---

## C    LIMITATIONS

Although the general log-augmented generation framework described in Algorithm 1 is compatible with any model during generation, the specific key-value (KV) log representation detailed in Section 2.2 and used in LAG is limited to open-source models. This is because accessing and modifying internal KV states is only possible with open models.

Furthermore, if each task a model encounters is vastly different from those it has previously handled—whether in domain, required knowledge, or other aspects—there may be little reusable reasoning. However, we believe this scenario is unlikely in practice, as it is common for users to pose similar questions repeatedly, either from the same individual or across different users.

## D    IMPLEMENTATION DETAILS

The experiments were conducted on a Linux cluster equipped with V100 and A100 GPUs. For model generation, we utilized Hugging Face (HF)'s `transformers` library and Llama-3.1-8B-Instruct hosted on HF. We also used the Snowflake embedding model hosted on HF, with embeddings computed using the `sentence_transformers` package.

The concatenated KV values of logs are passed to LLM using `past_key_values` parameter in the HuggingFace API: `model.generate(..., past_key_values=...)`.

## E    RESULTS ON DIFFERENT EMBEDDING MODELS

To assess the generalizability of LAG across different embedding models, we additionally evaluated it using the `Qwen3-Embedding-0.6B` retriever. As shown in Table 12, our method continues to deliver consistent performance gains.

Table 12: Performance using `Qwen3-Embedding-0.6B`.

|  | Musique | | MMLU-Pro | |
|---|---|---|---|---|
|  | EM | #Iterations ↓ | EM | #Iterations |
| Best baseline | 27.1 | 3.99 | 40.0 | 1.69 |
| LAG | **28.2** | **3.23** | **44.3** | **1.59** |

## F    RESULTS ON DIFFERENT LLMS

To evaluate the generalizability of LAG across different LLMs, we further tested it with `Qwen3-30B-A3B-Instruct-2507` and `Llama-3.1-70B-Instruct`. As shown in Table 13, LAG continues to outperform the strongest baseline.

Table 13: Performance using `Qwen3-30B-A3B-Instruct-2507` and `Llama-3.1-70B-Instruct`.

|  | Musique | | MMLU-Pro | |
|---|---|---|---|---|
|  | EM | #Iterations ↓ | EM | #Iterations |
| `Qwen3-30B-A3B-Instruct-2507` | | | | |
| Best baseline | 43.3 | 3.57 | 83.6 | 1.13 |
| LAG | **50.0** | **3.20** | **86.5** | **1.00** |
| `Llama-3.1-70B-Instruct` | | | | |
| Best baseline | 35.4 | 3.93 | 64.4 | 1.21 |
| LAG | **38.4** | **3.34** | **65.7** | **1.14** |

# G    RESULTS ON DYNAMIC LOG STORE

Table 14 presents results under the dynamic log store setting, showing that our method continues to deliver consistent performance gains.

Table 14: Performance under the dynamic log store setting.

|  | Musique | | MMLU-Pro | |
| --- | --- | --- | --- | --- |
|  | EM | #Iterations ↓ | EM | #Iterations |
| Best baseline | 24.3 | 3.67 | 40.0 | 1.69 |
| LAG | **25.6** | **2.88** | **42.2** | **1.54** |

# H    RESULTS ON DIFFERENT SPLITS

Table 15 reports results when X% of data is used to build the log store and Y% as unseen queries, under (X, Y) = (50, 50) and (30, 70). As in Section 3.2, LAG consistently outperforms the baselines on both knowledge- and reasoning-intensive datasets in terms of accuracy and efficiency.

Table 15: Exact Match (EM) and F1 scores on knowledge-intensive dataset Musique and reasoning-intensive dataset GPQA, along with the number of iterations required to complete each task. Seen or unseen refers to whether the questions were used to construct log stores. **Bolded** numbers indicate the best performance among all methods.

|  | Musique (seen) | | | Musique (unseen) | | | GPQA (seen) | | GPQA (unseen) | |
| --- | --- | --- | --- | --- | --- | --- | --- | --- | --- | --- |
|  | EM | F1 | #Iter. ↓ | EM | F1 | #Iter. | EM | #Iter. | EM | #Iter. |
| *X = 30, Y=70* | | | | | | | | | | |
| Standard agentic | 26.9 | 37.3 | 4.0 | 26.4 | 36.3 | 3.9 | 22.4 | 2.0 | 20.7 | 1.9 |
| LAG | **31.2** | **43.8** | **2.8** | **30.8** | **41.9** | **2.9** | **24.3** | **1.6** | **22.0** | **1.7** |
| *X = 50, Y=50* | | | | | | | | | | |
| Standard agentic | 25.9 | 36.4 | 3.9 | 27.1 | 37.0 | 4.0 | 21.9 | 2.0 | 20.6 | 1.9 |
| LAG | **30.8** | **42.5** | **2.7** | **31.5** | **42.7** | **2.8** | **24.1** | **1.6** | **25.0** | **1.6** |

## I  RESULTS WITH LIGHTWEIGHT SAFEGUARDS

Table 16 reports performance with the lightweight safeguards. Both thresholding and reflection substantially reduce negative transitions from correct to incorrect (C $\rightarrow$ I) and correct to unsolvable (C $\rightarrow$ U), and combining them yields the strongest robustness.

Table 16: Performance with lightweight safeguards. I, C, and U represent incorrect, correct, and unsolvable, respectively. X $\rightarrow$ Y refers to the number of questions that were initially X but became Y after incorporating logs.

|  | I $\rightarrow$ C | C $\rightarrow$ I | U $\rightarrow$ C | C $\rightarrow$ U | Total improvement |
|---|---|---|---|---|---|
| *Musique* | | | | | |
| LAG | +49 | -47 | +31 | -6 | +27 |
| LAG with reflection | +56 | -25 | +36 | -6 | +61 |
| LAG with threshold | +48 | -19 | +31 | -4 | +56 |
| LAG with reflection and threshold | +57 | -20 | +37 | -6 | +68 |
| *MMLU-Pro* | | | | | |
| LAG | +28 | -30 | +15 | -11 | +2 |
| LAG with reflection | +28 | -11 | +13 | -6 | +24 |
| LAG with threshold | +32 | -15 | +15 | -5 | +27 |
| LAG with reflection and threshold | +32 | -11 | +16 | -5 | +32 |

## J  THE USE OF LARGE LANGUAGE MODELS (LLMS)

LLM was used only to aid writing quality (proofreading and polishing grammar). No ideas, claims, methods, results, or references are generated by LLMs. All content decisions and revisions are made by the authors.

