# OpenReview forum: "Log-Augmented Generation: Scaling Test-Time Reasoning with Reusable Computation"
_ICLR.cc/2026/Conference — ICLR 2026 Poster_

### Official Review · Reviewer_JAu2 · 2025-10-15

**Soundness:** 3
**Presentation:** 4
**Contribution:** 4
**Rating:** 8
**Confidence:** 3

**Summary:**

This paper introduces Log-Augmented Generation (LAG), a framework designed to enable LLMs to reuse prior computations and reasoning traces at test time. The core technical innovation is to represent past execution logs not as text, but as key-value (KV) caches. The authors leverage the insight that a token's KV representation encapsulates the semantics of the entire preceding context. Their method encodes the full reasoning trace of a completed task into KV values but only stores the KV values corresponding to the final model response, creating a compact yet contextually rich representation. For new tasks, relevant logs are retrieved and their KV caches are injected into the model's generation process after adjusting for positional information. The paper evaluates LAG on both knowledge-intensive and reasoning-intensive QA datasets, demonstrating significant improvements in accuracy and efficiency over baselines, including standard agentic frameworks and reflection-based memory systems.

**Strengths:**

**Novel Use of KV Cache for Reasoning:** The paper's primary contribution is the repurposing of KV caching from a tool for computational efficiency to a mechanism for improving reasoning and accuracy by reusing past computations. This presents a compelling and conceptually distinct alternative to reflection-based memory systems.

*Effective Encoding Mechanism:** The technical approach of encoding the full reasoning history while only storing the KV values for the last model response is a clever solution to the trade-off between context richness and storage cost .

**Strong Empirical Performance:** The paper's claims are well-supported by a rigorous set of experiments with comprehensive baselines, including a no-memory agent, a text-based version of LAG, and prominent reflection-based systems. It demonstrates consistent and significant improvements across four challenging datasets spanning both knowledge- and reasoning-intensive tasks.

**Weaknesses:**

**Static Log Store Evaluation:** The experiments are conducted using a static log store that is built offline. This is a simplified setting that sidesteps critical challenges of a true lifelong learning system, such as the computational cost of retrieval in a massive and ever-growing log store, and the long-term effects of noise accumulation.

**Insufficient Analysis of Error Propagation:** The log store is intentionally populated without filtering for correctness to simulate a realistic scenario. While Table 3 shows a net positive impact, it also confirms that logs can mislead the model ($C \rightarrow I$) 7. The paper lacks a deeper qualitative or quantitative analysis of these failure cases and does not propose mechanisms to mitigate the risk of negative transfer from reusing flawed reasoning traces.

**Simple Retrieval Mechanism:** Retrieval is based on standard top-k semantic similarity 8. While effective in these experiments, this simple approach may become a bottleneck as the log store grows and the diversity of reasoning traces increases, potentially failing to retrieve the most relevant computational context.

**Questions:**

1. The experiments use a static log store. What new challenges would arise in a fully dynamic setting where new logs are added after every task? Specifically, how would the system manage retrieval efficiency and the risk of "polluting" the log store with noisy or incorrect reasoning over a long period?

2. Table 3 shows that logs can cause previously correct answers to become incorrect. Could you provide a qualitative example of this failure mode? What types of retrieved logs are most likely to mislead the model, and does this suggest a need for a more advanced mechanism to filter or weight retrieved logs based on some confidence metric?

3. The paper argues against abstraction because it can be lossy. However, abstraction also enables generalization. Your method seems optimized for tasks with significant sub-problem overlap. How do you expect LAG to perform on new tasks that are only distantly or conceptually related to problems in the log store, where direct computational reuse is not possible? Would an abstraction-based method potentially perform better in such a scenario?

---

> ### Author Response · Authors · 2025-12-04
> **Author response 1 (A1)**
>
> We thank the reviewer for the highly constructive and positive feedback. We appreciate that the reviewer recognizes the novelty of repurposing the KV Cache for reasoning and the cleverness of our encoding mechanism as a solution to the context-storage tradeoff. We hope the clarifications and additional context provided below reinforce the soundness of our method and help address the concerns raised.
>
> > W1: Static Log Store Evaluation: The experiments are conducted using a static log store that is built offline. This is a simplified setting that sidesteps critical challenges of a true lifelong learning system, such as the computational cost of retrieval in a massive and ever-growing log store, and the long-term effects of noise accumulation.
>
> > Q1: The experiments use a static log store. What new challenges would arise in a fully dynamic setting where new logs are added after every task? Specifically, how would the system manage retrieval efficiency and the risk of "polluting" the log store with noisy or incorrect reasoning over a long period?
>
> Thank you for the question. While our experiments use a static log store, we view the core challenges of large-scale retrieval and noise management as fundamentally similar in both static and dynamic settings. A large static store already exposes the system to the key issues of scale, noise, and interference.
>
> To assess this more directly, we also evaluated our approach in a dynamic log-store setting, with results shown in the table below. The results indicate that our method continues to deliver consistent performance gains, suggesting that it is effective for both static and dynamic log stores.
>
>
> | | Musique (Accuracy) | Musique (# Iterations) | MMLU-Pro (Accuracy) | MMLU-Pro (# Iterations) |
> |--------|------------------|-----------------------|-------------------|------------------------|
> | Best baseline | 24.3 | 3.67 | 40.0 | 1.69 |
> | LAG | **25.6** | **2.88** | **42.2** | **1.54** |

---

> ### Author Response · Authors · 2025-12-04
> **Author response 2 (A2)**
>
> > W2: Insufficient Analysis of Error Propagation: The log store is intentionally populated without filtering for correctness to simulate a realistic scenario. While Table 3 shows a net positive impact, it also confirms that logs can mislead the model ($C \rightarrow I$) 7. The paper lacks a deeper qualitative or quantitative analysis of these failure cases and does not propose mechanisms to mitigate the risk of negative transfer from reusing flawed reasoning traces.
>
> > W3: Simple Retrieval Mechanism: Retrieval is based on standard top-k semantic similarity 8. While effective in these experiments, this simple approach may become a bottleneck as the log store grows and the diversity of reasoning traces increases, potentially failing to retrieve the most relevant computational context.
>
> > Q2: Table 3 shows that logs can cause previously correct answers to become incorrect. Could you provide a qualitative example of this failure mode? What types of retrieved logs are most likely to mislead the model, and does this suggest a need for a more advanced mechanism to filter or weight retrieved logs based on some confidence metric?
>
> Thank you for raising these concerns. We observe that logs that are previously incorrect or irrelevant can introduce errors when solving new tasks. We agree that robustness to misleading or out-of-distribution logs is important, although our primary focus in this work is to demonstrate the benefits of log-augmented reasoning under a reasonable retrieval and prompting setup.
>
> To mitigate these cases, however, we further propose two lightweight safeguards that are straightforward to apply:
> * similarity thresholding: filters out low-relevance logs during retrieval
> * a brief reflection step: prompts the model to assess the reliability of the retrieved information before using it
>
> These mechanisms act as fallback strategies: when logs appear irrelevant or misleading, they are more likely to be filtered out or down-weighted, improving robustness on out-of-distribution tasks.
>
> The tables below summarize how each mechanism affects the transition of task outcomes (e.g., incorrect → correct, correct → incorrect, etc.) on representative knowledge-intensive (Musique) and reasoning-intensive (MMLU-Pro) datasets. As shown, the base LAG system occasionally exhibits harmful interference (e.g., correct → incorrect cases). However, both thresholding and reflection substantially reduce these negative transitions, and combining them yields the strongest robustness.
>
>
> | **Musique** | Incorrect → Correct | Correct → Incorrect | Unsolvable → Correct | Correct → Unsolvable | Total Improvement |
> |--------|-------------------|------------------|--------------------|--------------------|-----------------|
> | LAG | +49 | -47 | +31 | -6 | +27 |
> | LAG with reflection | +56 | -25 | +36 | -6 | +61 |
> | LAG with threshold | +48 | -19 | +31 | -4 | +56 |
> | LAG with both reflection and threshold | +57 | -20 | +37 | -6 | +68 |
>
> | **MMLU-Pro** | Incorrect → Correct | Correct → Incorrect | Unsolvable → Correct | Correct → Unsolvable | Total Improvement |
> |--------|-------------------|------------------|--------------------|--------------------|-----------------|
> | LAG | +28 | -30 | +15 | -11 | +2 |
> | LAG with reflection | +28 | -11 | +13 | -6 | +24 |
> | LAG with threshold | +32 | -15 | +15 | -5 | +27 |
> | LAG with both reflection and threshold | +32 | -11 | +16 | -5 | +32 |

---

> ### Author Response · Authors · 2025-12-04
> **Author response (A3)**
>
> > Q3: The paper argues against abstraction because it can be lossy. However, abstraction also enables generalization. Your method seems optimized for tasks with significant sub-problem overlap. How do you expect LAG to perform on new tasks that are only distantly or conceptually related to problems in the log store, where direct computational reuse is not possible? Would an abstraction-based method potentially perform better in such a scenario?
>
> We also evaluated LAG on tasks that are, as you note, only distantly or conceptually related to those in the log store: settings where direct computational reuse is limited and performance relies more on insight reuse. In such cases, previously stored reasoning traces can still provide value by suggesting effective strategies or helping the model avoid earlier mistakes.
>
> Table 11 in the paper provides a qualitative example demonstrating how our framework aids in solving a chemistry problem. The experimental section (Table 2 in the paper) further reports results on two reasoning-intensive benchmarks, GPQA and MMLU-Pro, which span general science and mathematics questions. Across these tasks, LAG consistently outperforms all baselines, indicating that it can generalize beyond settings with direct sub-problem overlap.
>
> We agree that abstraction could facilitate generalization in complementary ways. Exploring hybrid approaches that integrate LAG with abstraction-based methods is a promising direction for future work. As we have shown above in A2, even a lightweight reflective step during generation already enhances the effectiveness of our method.

---

### Official Review · Reviewer_TQuH · 2025-11-01

**Soundness:** 3
**Presentation:** 3
**Contribution:** 3
**Rating:** 6
**Confidence:** 4

**Summary:**

This paper introduces Log-Augmented Generation (LAG), a well-motivated and novel framework for enabling LLMs to reuse past computations by storing reasoning traces as KV caches. The key technical insight is to decouple the encoding context (the full reasoning trace) from the stored content (the KV values of the last response), which is clearly differentiated from prior work. This approach yields strong and consistent empirical improvements in both accuracy and efficiency over well-chosen baselines across multiple datasets. The experimental model in this paper is singular, lacking a formal analysis of the framework's reliability, generalization ability, or potential error propagation issues. Furthermore, the default method's storage costs are substantial, and the paper lacks an analysis of retrieval failures that lead to performance degradation.

**Strengths:**

1. **Clear and Motivated Problem Setting:** The paper identifies a practical and under-explored challenge—enabling LLMs to effectively reuse prior computation at test-time—mirroring a natural aspect of human reasoning. The distinction between reusing prior reasoning and simply increasing context length is well argued, particularly in the early discussion and Figure 1.
2. **Novel Use of** **KV** **Cache Representations:** Unlike existing KV cache approaches that target efficiency, LAG innovatively leverages the richer context encoded in KV values, advocating for selective storage (particularly from the last model response) to balance context richness and storage constraints. The distinction in Figure 2 between existing and proposed methods clearly demonstrates this key conceptual difference.
3. **Mathematical Clarity and Soundness:** The paper provides concrete mathematical details for handling positional encodings when reusing KV caches across different contexts, specifically for rotary embeddings. The approach to “stripping” and “reapplying” positional information is well formulated and critical to correctly reusing stored KV values.
4. **Comprehensive Experimental Evaluation:** Results in Tables 1 and 2 and subsequent ablations provide compelling evidence that LAG outperforms standard agentic, reflection, and existing KV cache methods in both accuracy (EM/F1) and efficiency (iterations), across diverse benchmarks. Qualitative examples and analyses further illuminate the mechanism by which LAG improves performance, especially via knowledge and insight reuse.
5. **Insightful Analysis:** The work dives into ablation over stored token count, the number of retrieved logs, and the tradeoff between accuracy and storage costs—which is rarely discussed rigorously in similar works.

**Weaknesses:**

1. **Related Work Positioning—Missing Direct Recent Papers:** The paper omits discussion of several key recent works that closely align with LAG's aims of scaling test-time computation and effective reuse for retrieval-augmented generation. Relevant missing works include:

   1. *Yue et al. (2025): Inference Scaling for Long-Context Retrieval Augmented Generation*.

   2. *Geiping et al. (2025): Scaling up Test-Time Compute with Latent Reasoning

       The absence of discussion or comparison with these is non-trivial: some address scaling and dynamic retrieval or propose closely related “augmented” or logic-based test-time reasoning, which could either complement or reveal subtle differences with LAG.

2. **Selection/Relevance Filtering Is Rudimentary:** Selection of logs uses standard semantic similarity on embeddings, but the method lacks adaptive or feedback-based techniques to mitigate harmful interference from irrelevant or misleading prior logs (see swings noted in **Table 3**, where some previously correct or unsolved problems become incorrect upon log incorporation). No mechanisms for detecting or counteracting negative transfer are proposed beyond manual thresholding.

3. **Lack of End-to-End Theoretical Analysis or Guarantees:** The framework is empirically strong, and equations for RoPE handling are rigorously presented, but the broader questions of reliability, generalization, or even error propagation with retrieved KV caches across very long log histories are not analyzed. There's no formal proof or error bound on when reused reasoning might be counterproductive.

4. The entire experimental validation is conducted on a **single model** (Llama-3.1-8B-Instruct). It is unknown how these findings generalize to other model architectures or, more importantly, to much larger models (e.g., 70B+). Larger models might be less reliant on logs, or their KV representations might behave differently.

**Questions:**

- **Cross-context** **KV** **Cache Reuse:** When reapplying RoPE to “stripped” KV values from logs, does the accumulated error remain negligible when these caches are reused repeatedly across highly diverse contexts or multiple retrieval rounds? Is there a scenario in which retrieved KV values harm rather than help due to drift or misalignment?
- **Task Drift and Out-of-domain Generalization:** How robust is LAG to highly dissimilar or out-of-distribution new tasks given log stores with only weakly relevant or misleading past traces? Are there any fallback mechanisms or metrics to assess when not to use retrieved logs?

---

> ### Author Response · Authors · 2025-12-04
> **Author response (A1)**
>
> We thank the reviewer for the thoughtful and highly positive feedback, which strongly validates the core contributions of our work, particularly the clear problem setting, the novel use of KV cache representations, and the mathematical clarity of our approach. We appreciate that the reviewer found the comprehensive experimental evaluation compelling and the analysis insightful. We hope the clarifications and additional context provided below reinforce the soundness of our method and help address the concerns raised.
>
> > W1: Related Work Positioning—Missing Direct Recent Papers: The paper omits discussion of several key recent works that closely align with LAG's aims of scaling test-time computation and effective reuse for retrieval-augmented generation.
>
> Thank you for pointing out these relevant works. We will include and discuss them in the next version of the paper. While these papers focus on test-time scaling through text-based iterative decompositions or training methods, our approach centers on retrieving past reasoning traces and leveraging KV-based encoding and storage. We believe both aspects are novel and distinguish our method from prior work.

---

> ### Author Response · Authors · 2025-12-04
> **Author response (A2)**
>
> > W2: Selection/Relevance Filtering Is Rudimentary: Selection of logs uses standard semantic similarity on embeddings, but the method lacks adaptive or feedback-based techniques to mitigate harmful interference from irrelevant or misleading prior logs (see swings noted in Table 3, where some previously correct or unsolved problems become incorrect upon log incorporation). No mechanisms for detecting or counteracting negative transfer are proposed beyond manual thresholding.
>
> > W3: Lack of End-to-End Theoretical Analysis or Guarantees: The framework is empirically strong, and equations for RoPE handling are rigorously presented, but the broader questions of reliability, generalization, or even error propagation with retrieved KV caches across very long log histories are not analyzed. There's no formal proof or error bound on when reused reasoning might be counterproductive.
>
> > Q2: Task Drift and Out-of-domain Generalization: How robust is LAG to highly dissimilar or out-of-distribution new tasks given log stores with only weakly relevant or misleading past traces? Are there any fallback mechanisms or metrics to assess when not to use retrieved logs?
>
> Thank you for raising these concerns. We agree that robustness to misleading or out-of-distribution logs is important, although our primary focus in this work is to demonstrate the benefits of log-augmented reasoning under a reasonable retrieval and prompting setup.
>
> To mitigate these cases, however, we further propose two lightweight safeguards that are straightforward to apply:
> * similarity thresholding: filters out low-relevance logs during retrieval
> * a brief reflection step: prompts the model to assess the reliability of the retrieved information before using it
>
> These mechanisms act as fallback strategies: when logs appear irrelevant or misleading, they are more likely to be filtered out or down-weighted, improving robustness on out-of-distribution tasks.
>
> The tables below summarize how each mechanism affects the transition of task outcomes (e.g., incorrect → correct, correct → incorrect, etc.) on representative knowledge-intensive (Musique) and reasoning-intensive (MMLU-Pro) datasets. As shown, the base LAG system occasionally exhibits harmful interference (e.g., correct → incorrect cases). However, both thresholding and reflection substantially reduce these negative transitions, and combining them yields the strongest robustness.
>
>
> | **Musique** | Incorrect → Correct | Correct → Incorrect | Unsolvable → Correct | Correct → Unsolvable | Total Improvement |
> |--------|-------------------|------------------|--------------------|--------------------|-----------------|
> | LAG | +49 | -47 | +31 | -6 | +27 |
> | LAG with reflection | +56 | -25 | +36 | -6 | +61 |
> | LAG with threshold | +48 | -19 | +31 | -4 | +56 |
> | LAG with both reflection and threshold | +57 | -20 | +37 | -6 | +68 |
>
> | **MMLU-Pro** | Incorrect → Correct | Correct → Incorrect | Unsolvable → Correct | Correct → Unsolvable | Total Improvement |
> |--------|-------------------|------------------|--------------------|--------------------|-----------------|
> | LAG | +28 | -30 | +15 | -11 | +2 |
> | LAG with reflection | +28 | -11 | +13 | -6 | +24 |
> | LAG with threshold | +32 | -15 | +15 | -5 | +27 |
> | LAG with both reflection and threshold | +32 | -11 | +16 | -5 | +32 |

---

> ### Author Response · Authors · 2025-12-04
> **Author response (A3)**
>
> > W4: The entire experimental validation is conducted on a single model (Llama-3.1-8B-Instruct). It is unknown how these findings generalize to other model architectures or, more importantly, to much larger models (e.g., 70B+). Larger models might be less reliant on logs, or their KV representations might behave differently.
>
> To further assess the generalizability of our approach across models with larger context windows, higher parameter counts, and different architectures, we additionally evaluated `Qwen3-30B-A3B-Instruct-2507` (30B parameters, 262K context length) and `Llama-3.1-70B-Instruct` (70B parameters, 128K context length). For comparison, `Llama-3.1-8B-Instruct`, used in the paper, has 8B parameters and a 128K context length. As shown in the table below, our method consistently outperforms the strongest baseline across all models, demonstrating its robustness and effectiveness even as context size and model scale vary.
>
>
> |  | Method | Musique (Accuracy) | Musique (# Iterations) | MMLU-Pro (Accuracy) | MMLU-Pro (# Iterations) |
> |------------------------|--------|------------------|-----------------------|-------------------|------------------------|
> | `Qwen3-30B-Instruct` (262K context length) | Best baseline | 43.3 | 3.57 | 83.6 | 1.13 |
> | `Qwen3-30B-Instruct` (262K context length) | LAG | **50.0** | **3.20** | **86.5** | **1.00** |
> |  |
> | `Llama-3.1-70B-Instruct` (128K context length) | Best baseline | 35.4 | 3.93 | 64.4 | 1.21 |
> | `Llama-3.1-70B-Instruct` (128K context length) | LAG | **38.4** | **3.34** | **65.7** | **1.14** |

---

> ### Author Response · Authors · 2025-12-04
> **Author response (A4)**
>
> > Q1: Cross-context KV Cache Reuse: When reapplying RoPE to “stripped” KV values from logs, does the accumulated error remain negligible when these caches are reused repeatedly across highly diverse contexts or multiple retrieval rounds? Is there a scenario in which retrieved KV values harm rather than help due to drift or misalignment?
>
> We allow KV values to be reused across multiple rounds, and we described mechanisms in A2 (including similarity thresholding and brief reflection) that help mitigate potential errors introduced by past KV values. These measures aim to reduce drift or misalignment and limit scenarios in which retrieved caches could harm performance.

---

### Official Review · Reviewer_QLNm · 2025-11-03

**Soundness:** 3
**Presentation:** 2
**Contribution:** 2
**Rating:** 4
**Confidence:** 3

**Summary:**

This paper presents "LAG", Log-Augmented generation, which modifies standard RAG procedures to use logs/generated reasoning sequences. Furthermore, it uses KV-cache intermediate values as the retrieved representations. The paper presents strong empirical evidences for the method proposed.

**Strengths:**

* Use of retrieval for reasoning is very nice and the idea of using intermediate KV cache representations. This is a very nice idea and I believe an important area of study as it helps to decouple knowledge representation and reasoning representations.
* The empirical successes of the paper especially using the KV cache values rather than full text representation of logs motivates future exploration of a wide variety of topics in retrieval augmented methods.

**Weaknesses:**

* **Scaling with Context Length**:  The KV cache vs text result is most interesting I believe, but also under explored. It is of course related to context length limitations, but it is hard to know exactly how the context size of the model changes this (e.g., Llama 8B's performance: https://arxiv.org/pdf/2504.06214v1).
* **Depth of contribution**: It's not clear to me how to evaluate the novelty of the contribution here. I understand the novelty of retrieving logs, but it is not a sea-change from past work that retrieved data (e.g., https://arxiv.org/pdf/2203.08773 or https://arxiv.org/pdf/2311.07850).
* **Overall Metric Scores**: It's perhaps hard to evaluate the quality of the method, when as I understand the metrics for GPQA are pretty far off of SOTA?

**Questions:**

1. What is the context length for the retrieval model? (Sorry If I missed this) What is the length of the logs retrieved?
2. Given the logs are likely OOD for the retrieval model, why do you think you see such nice retrieval performance? Is it the quality of snowflake in general? Or a characteristic of the data?
3. What exactly is the query of the retrieval step? Formatting, context, instruction, etc?

---

> ### Author Response · Authors · 2025-12-04
> **Author response 1 (A1)**
>
> We thank the reviewer for the thoughtful feedback. We appreciate that the idea of leveraging retrieval for reasoning, as well as using intermediate KV cache representations, resonated as a meaningful contribution. We also appreciate the recognition of our empirical results, particularly the effectiveness of using KV cache values instead of full text representations, and agree that this opens promising directions for further exploration in retrieval-augmented methods. We hope the clarifications and additional context provided below reinforce the soundness of our method and help address the concerns raised.
>
> > W1: Scaling with Context Length: The KV cache vs text result is most interesting I believe, but also under explored. It is of course related to context length limitations, but it is hard to know exactly how the context size of the model changes this (e.g., Llama 8B's performance: https://arxiv.org/pdf/2504.06214v1).
>
> To further assess the generalizability of our approach across models with larger context windows, higher parameter counts, and different architectures, we additionally evaluated `Qwen3-30B-A3B-Instruct-2507` (30B parameters, 262K context length) and `Llama-3.1-70B-Instruct` (70B parameters, 128K context length). For comparison, `Llama-3.1-8B-Instruct`, used in the paper, has 8B parameters and a 128K context length. As shown in the table below, our method consistently outperforms the strongest text-only baseline across all models, demonstrating its robustness and effectiveness even as context size and model scale vary.
>
>
> |  | Method | Musique (Accuracy) | Musique (# Iterations) | MMLU-Pro (Accuracy) | MMLU-Pro (# Iterations) |
> |------------------------|--------|------------------|-----------------------|-------------------|------------------------|
> | `Qwen3-30B-Instruct` (262K context length) | Best text baseline | 43.3 | 3.57 | 83.6 | 1.13 |
> | `Qwen3-30B-Instruct` (262K context length) | LAG | **50.0** | **3.20** | **86.5** | **1.00** |
> |  |
> | `Llama-3.1-70B-Instruct` (128K context length) | Best text baseline | 35.4 | 3.93 | 64.4 | 1.21 |
> | `Llama-3.1-70B-Instruct` (128K context length) | LAG | **38.4** | **3.34** | **65.7** | **1.14** |

---

> ### Author Response · Authors · 2025-12-04
> **Author response 2 (A2)**
>
> > W2: Depth of contribution: It's not clear to me how to evaluate the novelty of the contribution here. I understand the novelty of retrieving logs, but it is not a sea-change from past work that retrieved data (e.g., https://arxiv.org/pdf/2203.08773 or https://arxiv.org/pdf/2311.07850).
>
> We would like to clarify the two aspects of our contribution that go beyond prior work, including the papers cited:
> * **A new representation of past experience via KV-based reasoning traces**: While it is true that our method learns from past logs, our contribution lies in how past experience is represented. Prior retrieval-augmented methods, including those cited, store past information as text or graph structures, which typically require manual curation, formatting, or filtering. In contrast, we introduce KV vectors generated naturally during model decoding as the representation of past experience. These traces preserve rich contextual and intermediate reasoning information “for free,” without additional hand-crafted preprocessing, and reflect a qualitatively different source of signal than curated textual data. Our experiments show that these KV traces are both scalable to collect and empirically useful.
> * **A complete system that makes retrieval and reuse of KV traces feasible for downstream tasks**: We also contribute the first unified framework that makes this representation practically useful: (i) selecting and storing KV traces, (ii) retrieving them via semantic similarity, and (iii) re-integrating them into the decoding process with positional re-encoding. We show that this end-to-end pipeline leads to consistent improvements on knowledge-intensive and reasoning benchmarks.
>
> Together, these contributions extend retrieval-from-logs beyond prior text-based approaches by introducing a new, automatically obtained representation of past experience and the system needed to exploit it effectively.

---

> ### Author Response · Authors · 2025-12-04
> **Author response 3 (A3)**
>
> > W3: Overall Metric Scores: It's perhaps hard to evaluate the quality of the method, when as I understand the metrics for GPQA are pretty far off of SOTA?
>
> GPQA scores can vary significantly depending on the underlying LLMs, prompting strategies, and retrieval components. Our goal in this work is to validate the effectiveness of our method under a controlled, reasonable baseline setup. By keeping the model and prompt strategies fixed across all experimental setups, we ensure a fair apples-to-apples comparison. Importantly, the absolute scores we obtain are fully aligned with publicly reported performance for the same model. With Llama-3.1-8B-Instruct, our method reaches ~25–30% accuracy on GPQA, which is comparable to, if not higher than, the 25.9% accuracy reported on the GPQA leaderboard for this model (https://artificialanalysis.ai/evaluations/gpqa-diamond?models=llama-3-1-instruct-8b). This alignment gives us confidence that (1) our evaluation setup is sound, and (2) the improvements we observe over baselines genuinely reflect the effectiveness of our method rather than experimental artifacts.

---

> ### Author Response · Authors · 2025-12-04
> **Author response 4 (A4)**
>
> > Q1: What is the context length for the retrieval model? (Sorry If I missed this) What is the length of the logs retrieved?
>
> The retriever model has a context length of 8192 tokens. Each log entry stored in the log store (and thus retrieved) is approximately 232 tokens on average, and at each step of the agentic reasoning loop we allow up to three new logs to be retrieved.
>
> > Q3: What exactly is the query of the retrieval step? Formatting, context, instruction, etc?
>
> The query used at each retrieval step is the task description generated by the model itself—that is, the subtask the model identifies as potentially helpful for solving the original problem. Retrieval is performed over two sources: (1) the document corpus (when available, such as in open-domain QA) and (2) the log store containing past reasoning traces. We retrieve the top-k items with the highest semantic similarity to the model-proposed query from both sources. All details regarding formatting, context construction, and instructions for how the retrieved content is incorporated into model generation are provided in Appendix A of the paper.

---

> ### Author Response · Authors · 2025-12-04
> **Author response 5 (A5)**
>
> > Q2: Given the logs are likely OOD for the retrieval model, why do you think you see such nice retrieval performance? Is it the quality of snowflake in general? Or a characteristic of the data?
>
> Our retrieval works by matching the semantic similarity between the new task and the reasoning traces stored in the log store. Because the retriever’s role is to align *meanings* rather than to model the distribution of the logs themselves, its effectiveness does not depend on the logs being in-distribution. In practice, this is similar to matching a question to a relevant (partial) answer, a task that embedding models are commonly trained to capture. For this reason, we expect the retrieval performance to remain robust across different retrieval models, independent of the specific characteristics of `snowflake-arctic-embed-m-v2.0` or the particular distribution of the logs.
>
> To examine this more directly, we evaluated an additional retriever, `Qwen/Qwen3-Embedding-0.6B`, and report the results in the following table. The tables show that our method continues to yield consistent performance improvements. This suggests that the gains are not tied to a particular retriever (e.g., `snowflake-arctic-embed-m-v2.0`) but instead stem from characteristics of the reasoning-trace data itself and the way our framework leverages it.
>
> | | Musique (Accuracy) | Musique (# Iterations) | MMLU-Pro (Accuracy) | MMLU-Pro (# Iterations) |
> |--------|------------------|-----------------------|-------------------|------------------------|
> | Best baseline | 27.1 | 3.99 | 40.0 | 1.69 |
> | LAG | **28.2** | **3.23** | **44.3** | **1.59** |

---

### Author Response · Authors · 2025-12-04
**Rebuttal summary**

We thank the reviewers for their thoughtful and constructive feedback.

Besides treating past reasoning traces as a new information source, we highlight additional contributions compared to prior work:
* **A new representation of past experience via KV-based reasoning traces**: Prior retrieval-augmented methods store past information as text or graph structures, which typically require manual curation, formatting, or filtering. In contrast, we introduce KV vectors generated naturally during model decoding as the representation of past experience. These traces preserve rich contextual and intermediate reasoning information “for free,” without additional hand-crafted preprocessing, and reflect a qualitatively different source of signal than curated textual data. Our experiments show that these KV traces are both scalable to collect and empirically useful.
* **A complete system that makes retrieval and reuse of KV traces feasible for downstream tasks**: We also contribute the first unified framework that makes this representation practically useful: (i) selecting and storing KV traces, (ii) retrieving them via semantic similarity, and (iii) re-integrating them into the decoding process with positional re-encoding. We show that this end-to-end pipeline leads to consistent improvements on knowledge-intensive and reasoning benchmarks.

All reviewers appreciate the paper's novel use of KV cache representations for reasoning, calling it a significant advance that efficiently reuses prior computation while effectively decoupling knowledge and reasoning. They highlighted the clever technique of selectively storing KV values to balance context richness and storage constraints. `Reviewer TQuH` also specifically noted the clear problem setting and the mathematical clarity in handling positional encodings during generation. All reviewers agreed that the strong empirical performance and comprehensive evaluation across diverse benchmarks compellingly validated the paper's claims of consistent improvements in accuracy and efficiency.


`Reviewer QLNm` raises W1-W3 and Q1-Q3.

| Summary of reviews | Summary of author response |
|--------------------|----------------|
| **W1:** The generalizability of our framework to larger models and longer context windows beyond the `Llama-3.1-8B-Instruct` (128K context) model used in the paper. | **A1:** We evaluated additional models with different architectures and substantially larger context lengths, including `Qwen3-30B-Instruct` (262K context) and the larger `Llama-3.1-70B-Instruct`. On both a representative knowledge-intensive dataset (Musique) and a reasoning-intensive dataset (MMLU-Pro), we continue to outperform the strongest baselines, demonstrating the robustness of our approach. |
| **W2:** How does LAG differ from prior data retrieval work? | **A2:** We clarified that prior retrieval-augmented methods primarily rely on curated textual/graph logs. In contrast, our method introduces KV-based reasoning traces, automatically generated during decoding and capturing rich intermediate reasoning, as a new representation of past experience. We also present the first full framework for selecting, retrieving, and reintegrating these traces into model decoding. This novel representation, together with the end-to-end system built to exploit it, sets our approach apart from existing work. |
| **W3:** The interpretability of the performance of LAG given that GPQA scores are below state-of-the-art benchmarks. | **A3:** We clarified that GPQA scores depend on the model and setup, and our controlled evaluation with `Llama-3.1-8B-Instruct` matches publicly reported performance, ensuring that improvements over baselines reflect the method’s effectiveness rather than experimental artifacts. |
| **Q1, Q3:** Clarification on the retrieval model’s context length, retrieved log sizes, and the formulation of retrieval queries. | **A4:** We specified the model’s context length, average log size, and explained the retrieval query and formatting details in Appendix A. |
| **Q2:** The robustness of our method to different choices of embedding models. | **A5:** We additionally reported results using `Qwen3-Embedding-0.6B`, a widely used embedding model distinct from the `snowflake-arctic-embed-m-v2.0` model used in the main paper, and confirmed that our method remains effective. |

---

> ### Author Response · Authors · 2025-12-04
> **Rebuttal summary (continued)**
>
> `Reviewer TQuH` raises W1-W4 and Q1-Q2.
>
> | Summary of reviews | Summary of author response |
> |-----------------|----------------|
> | **W1:** How does LAG differ from prior work on scaling test-time computation and effective reuse? | **A1:** We clarified that prior test-time scaling and retrieval-augmented methods primarily rely on text-based decompositions or training methods. In contrast, our method centers on KV-based reasoning traces, automatically generated during decoding and capturing rich intermediate reasoning, as a new representation of past experience. We also present the first full framework for selecting, retrieving, and reintegrating these traces into model decoding. This novel representation, together with the end-to-end system built to exploit it, sets our approach apart from existing work. |
> | **W2, W3, Q2:** How does LAG deal with incorrect or misleading logs? | **A2:** We introduced two lightweight safeguards: a similarity threshold that filters out low-relevance logs during retrieval, and a brief reflection step that prompts the model to evaluate the reliability of retrieved information before using it. Our experiments show that incorporating these strategies substantially mitigates the negative impact of incorrect or misleading logs. |
> | **W4:** The generalizability of our framework to larger models and longer context windows beyond the `Llama-3.1-8B-Instruct` (128K context) model used in the paper | **A3:** We evaluated additional models with different architectures and substantially larger context lengths, including `Qwen3-30B-Instruct` (262K context) and the larger `Llama-3.1-70B-Instruct`. On both a representative knowledge-intensive dataset (Musique) and a reasoning-intensive dataset (MMLU-Pro), we continue to outperform the strongest baselines, demonstrating the robustness of our approach. |
> | **Q1:** Potential risks of error accumulation or misalignment when reusing KV caches across diverse contexts or multiple retrieval rounds. | **A4:** We described mechanisms like similarity thresholding and brief reflection to mitigate drift (A2), reducing the likelihood that retrieved KV caches harm performance. |
>
> `Reviewer JAu2` raises W1-W3 and Q1-Q3.
>
> | Summary of reviews | Summary of author response |
> |-----------------|----------------|
> | **W1, Q1:** How does LAG generalize and handle challenges in a dynamically updated log store compared to the static, offline-constructed setting? | **A1:** We clarified that both static and dynamic settings exhibit similar challenges. We further conducted experiments under a dynamic log-store setting on both Musique and MMLU-Pro. The results again show that we surpass the best baselines, supporting the effectiveness of our approach. |
> | **W2, W3, Q2:** How does LAG deal with incorrect or misleading logs? | **A2:** We introduced two lightweight safeguards: a similarity threshold that filters out low-relevance logs during retrieval, and a brief reflection step that prompts the model to evaluate the reliability of retrieved information before using it. Our experiments show that incorporating these strategies substantially mitigates the negative impact of incorrect or misleading logs. |
> | **Q3:** How does LAG handle tasks only distantly related to those in the log store, and whether abstraction might help? | **A3:** We clarified that LAG was evaluated on reasoning-intensive tasks only distantly related to the log store, including GPQA and MMLU-Pro, where it consistently outperforms baselines. We agree that abstraction could help since lightweight reflection can enhance performance as discussed in A2. |
>
> Overall, we believe our responses address the reviewers’ concerns regarding clarity, generalizability, and error handling. We hope these clarifications highlight the technical novelty, practical relevance, and strength of our work.

---

### Meta-Review · Area_Chair_8kTP · 2026-01-06

**Summary:**

This work studies the acceleration of LLM inference. The authors propose a method that reuses KV caches from previous generations across different tasks, thereby avoiding redundant recomputation when new tasks share overlapping content with prior ones. Although the core idea is simple, the experimental results appear promising. Reviewers raised concerns about the limited experimental scope, as evaluations are conducted only on the LLaMA-3 8B model, as well as the absence of theoretical guarantees. I would encourage the authors to include experiments across a wider range of model scales to better demonstrate the generality of the proposed approach.

**Reviewer Concerns:**

**Unaddressed concerns:**

**Reviewer QLNm:** none.

**Reviewer TQuH:** lack of theoretical guarantees and insufficient experimental evaluation.

**Reviewer JAu2:** none.

**Reviewer Scores:**

**Reviewer QLNm:** 4 → 6

**Reviewer TQuH:** 6 → 6

**Reviewer JAu2:** 8 → 8

---

### Decision · Program_Chairs · 2026-01-26

Accept (Poster)